# Function regression using the forward forward training and inferring paradigm

## ABSTRACT

Function regression/approximation is a fundamental application of machine learning. Neural networks (NNs) can be easily trained for function regression using a sufficient number of neurons and epochs. The forward-forward learning algorithm is a novel approach for training neural networks without backpropagation, and is well suited for implementation in neuromorphic computing and physical analogs for neural networks. To the best of the authors' knowledge, the Forward Forward paradigm of training and inferencing NNs is currently only restricted to classification tasks. This paper introduces a new methodology for approximating functions (function regression) using the Forward-Forward algorithm. The paper further evaluates the developed methodology on univariate and multivariate functions and benchmarks the framework on open source regression data, while comparing its performance to other regression techniques.

## 1 INTRODUCTION

The computational demands associated with the training and inference of AI models result in substantial energy consumption. As the adoption of AI continues to grow exponentially, the associated escalating energy requirement also poses significant challenges, necessitating the development of energy-efficient alternatives for computational processes (Mehonic & Kenyon, 2022). Brain-inspired (Neuromorphic) computing paradigms are designed to mimic the brain's computing processes, in order to translate the energy efficiency of the neural connections in the brain to computing tasks. Other analog or physical systems can also be considered useful tools for developing energy-efficient solutions for computing (Zolfagharinejad et al., 2024). An example of a physical computing device would be the memristor called "dot-product engine" (Li et al., 2023; Zhang et al., 2020; Chen et al., 2023). It serves as a physical analog for matrix-vector multiplication and performs the multiplication in a single step instead of the usual $n^2$ steps in traditional computing methods (Sharma et al., 2024).

Neural Networks (NNs) are fundamental building blocks to deep learning frameworks. Their implementation on classical digital computers is energy-intensive. To address this high energy consumption, recent trends have focused on exploring physical systems that may serve as analogs to digital neural networks (Wright et al., 2022). These are popularly called physical neural networks. Physical neural networks use physical systems or materials to emulate the behavior of neurons and synapses. Neural networks use complex activation functions to add non-linearity to the system. Similarly, these physical systems use physical phenomena that help surrogate these activation functions and layer-wise computations. Recent works have demonstrated that using physical systems for neural network computations is not only highly energy-efficient but also achieves above 90% classification accuracy (Wright et al., 2022; Momeni et al., 2023a).

Training of neural networks is popularly done using the backpropagation algorithm (Linnainmaa, 1970; 1976; Griewank, 2012). The backpropagation (BP) algorithm uses supervised learning to optimize the loss function using methods like stochastic gradient descent. However, the BP algorithm is highly energy inefficient because of the need for forward and backward passes for each step of the optimization. The backward pass calculates the gradient of the loss function with respect to each parameter in the network. This requires a lot of energy and time, which only increases with increase in depth and complexity of the NN architecture. In addition, backpropagation-based parameter learning is not suitable for multi-layered physical neural networks which are unidirectional in

time. Earlier works implementing physical neural networks (Wright et al., 2022) used digital twins of the physical system to achieve backpropagation during training. However, digital twins add to the energy cost during the training process and are not accurately available for many physical systems. Therefore, the ability to train NNs without BP has the potential to significantly improve the energy efficiency of the aforementioned physical NNs.

In 2022, Hinton (2022) proposed a new algorithm called the Forward-Forward (FF) algorithm, which uses only a forward pass to train the neural networks. This neuromorphic algorithm is based on the idea of learning by comparing correctly labeled and incorrectly labeled data. As the FF algorithm is unidirectional, it can be used to train physical neural networks. Layer-wise training in this algorithm trains each layer of the network to correctly distinguish between correctly labeled and incorrectly labeled data points. Subsequent to the introduction of the FF algorithm, various researchers have extended it to various applications like convolutional neural networks (Scodellaro et al., 2023), recurrent neural networks (Kag & Saligrama, 2021), etc. Some have also developed variations of this algorithm to enhance it for better accuracy (Wu et al., 2024; Lorberbom et al., 2024). Some researchers have also successfully implemented the FF approach to train neural networks using physical systems combined with digital linear layers (Momeni et al., 2023a). The FF Algorithm is designed to solve classification problems and to the best of the authors' knowledge all of the prior works only address classification tasks. In this work, we seek to extend the FF algorithm for the function regression task. In section 2 we discuss the theoretical background of the FF algorithm and how it can be extended to regression tasks, with section 3 discussing the 1-D, 2-D, 3-D function benchmarks, and three open source dataset benchmarks for validating the proposed FF-regression algorithm, and section 4 concluding the manuscript.

## 2 THEORETICAL BACKGROUND

### 2.1 FORWARD FORWARD TRAINING

The Forward-Forward algorithm is an approach to train neural networks layer-wise without using backpropagation, by relying on learning by comparison. This is developed from the idea of contrastive learning, where both correct and incorrect data are required for training the model by comparison (Khosla et al., 2020). This algorithm takes two types of data– correctly labeled (positive) and incorrectly labeled (negative) data. A function called "goodness" quantifies each layer's output. The input for the goodness function of a layer is the layer's output. The output of the layer's goodness function is a scalar value. The goodness value for positive and negative datasets is calculated separately. The weights of the layer are optimized to maximize the difference between the goodness for the positive data and the goodness for the negative data. Thus, each layer in the network is optimized to discriminate between positive (correctly labeled) and negative (incorrectly labelled) data. Furthermore, this is done serially (layer-by-layer) and independently without any backward propagation of gradients. A schematic of a NN trained and inferred using the FF paradigm is provided in Fig. 1a.

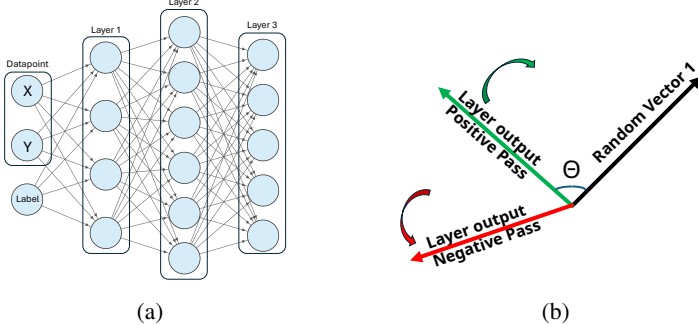

(a)                                                                (b)

Figure 1: (a) Schematic diagram of neural network which can be trained using forward forward algorithm. Note that the last layer is also the same as a hidden layer, i.e., there is no output from the final layer. (b) Arbitrary vectors used to optimize the difference between positive and negative goodness.

## 2.2 Discussion on Goodness function

The goodness function can be any function that receives the outputs of a NN layer as an argument and returns a scalar value. A higher value of goodness for a layer indicates that the datapoint-label combination are correct (correctly labeled), while a lower value of goodness indicates that the combination is incorrect (incorrectly labeled). In Hinton (2022), it was suggested to define goodness function as the sum of squares of layer output. The outputs of a layer are then normalized before being passed as inputs to the next layer in order to avoid biasing subsequent layers. Another more recent work (Momeni et al., 2023a) used cosine similarity as a goodness function. The cosine similarity is evaluated between a layer output and a fixed random vector of equal dimension. As illustrated in Fig. 1b, the objective for training the layer is to increase the cosine similarity between the layer output and the vector for positive data and to reduce it for negative data. This training increases the gap between goodness associated with positive data (correctly labeled) and negative data (incorrectly labeled). Note that arbitrary vectors should be different for different layers of the neural network. The particular definition of the goodness function adopted in Momeni et al. (2023a) has an added advantage of not requiring renormalization of layer outputs at every layer. In the current work, the goodness function used in Momeni et al. (2023a) is adopted.

## 2.3 Discussion on loss function and training

During training, the dataset is categorized into positive (correctly labeled) and negative (incorrectly labeled). The goodness associated with the positive and negative data for the $i^{th}$ layer is written as in Eqs. 1 and 2:

$$g_{\text{pos}}^{(i)} = cos_{sim}(y_{\theta_{(i)},\text{pos}}^{(i)}, \zeta^{(i)}) \tag{1}$$

$$g_{\text{neg}}^{(i)} = cos_{sim}(y_{\theta_{(i)},\text{neg}}^{(i)}, \zeta^{(i)}) \tag{2}$$

where, $g_{\text{pos}}^{(i)}, g_{\text{neg}}^{(i)} \in \mathbb{R}$ are the goodness values of positive data and negative data, respectively, associated with the outputs $y_{\text{pos},\theta_{(i)}}^{(i)}, y_{\text{neg},\theta_{(i)}}^{(i)} \in \mathbb{R}^{d_i}$ of layer $i$. $\zeta^{(i)} \in \mathbb{R}^{d_i}$ is a fixed arbitrary vector of dimension $d_i$. The goodnesses are evaluated as the cosine similarity ($cos_{sim}$) between the vectors $y_{\theta_{(i)}}^{(i)}$ and $\zeta^{(i)}$.

Given that the training is performed layer-wise, the Loss function is defined for each layer. As discussed in section 2.2, the minimization of the layer's loss function should lead to the maximization of the difference between $g_{\text{pos}}$ and $g_{\text{neg}}$. Following Momeni et al. (2023b) the layer-wise loss function used in the current work is given in Eq. 3

$$Loss^{(i)} = \log(1 + \exp(-\theta\delta^{(i)})) \tag{3}$$

where $\delta^{(i)} = g_{\text{pos}}^{(i)} - g_{\text{neg}}^{(i)}$ From Eq. 3 it is evident that the layer's loss is minimized by maximizing $\delta^{(i)}$, i.e, by maximizing $g_{\text{pos}}$ and minimizing $g_{\text{neg}}$.

## 2.4 Inferencing in Forward Forward Neural Networks

Regular NNs trained using the BP paradigm have neural connections that flow between the input neuron and the output neuron(s). However, as seen in Fig. 1a, neural networks using the FF paradigm do not have such input-to-output neural connections. Instead, the "input" data point X and the "output" data point Y both feature in the input neural layer alongside the label L, i.e, the classification label for the data point is encoded as a part of the input to the neural network. As discussed in the previous sections, each layer of the Forward-Forward NNs are trained to provide low goodness outputs in cases where the datapoint-label combination is incorrect, and provide high goodness outputs for correct datapoint-label combinations. During forward inferencing, the label which provides the highest sum of goodness outputs across all layers is selected as the "correct" label corresponding to the data point. This implies that the forward pass must be performed for all labels for a given data point in order to identify the correct label/classification for the given data. In comparison, NNs trained using BP would require a single forward pass during inferencing.

## 2.5 FUNCTION REGRESSION USING FORWARD FORWARD APPROACH

As discussed in the preceding sections, the FF algorithm is primarily suited to classification tasks wherein the input datapoint-label combination is classified as either "correct" or "incorrect" by the NN. In this context, we can view the task of function regression as classification of datapoints as either within a preset tolerance level of the training data, or outside of the preset tolerance level of the training data. The case of training a FF NN to approximate a 1-D function is illustrated in figure 2, where the training points are highlighted as blue crosses, and the trial points are the colored circles. The errorbars highlight the user-defined tolerance level (tol), within which a trial point is considered to belong to the function-value (in-tol). All trial points outside the errorbar are considered to be not equal to the function-value (out-tol). All in-tol points are colored in green and out-tol points are colored in red for visual illustration. Next, we chose a labelling scheme that assigns the label 1 to in-tol points and 0 to out-tol points. Thus, to train a FF NN, the positive (correctly labeled) dataset will consist of 1 assigned to in-tol points and 0 to out-tol points and the negative dataset will have 1 assigned to out-tol points and 0 assigned to in-tol points. As shown in figure 1a, the inputs to training a FF NN would be the x and y co-ordinate of the trial points and the associated labels. Each layer in the FF NN would be trained to minimize Eq. 3, i.e., maximize the difference in goodness output between the positive and negative dataset. Thus, a well-trained FF NN is expected to classify any given point in space as either in-tol and out-tol. Algorithm 1 summarizes the training of a FF NN for regression.

We make use of this ability of the FF NN to discriminate between out-tol and in-tol points to obtain the mean value and standard deviation of the function at a point that is not in the training data-set. This forward inferencing of the FF NN for regression is illustrated in figure 3, wherein the value of the function is obtained at $x_{\text{query}}$ ($\notin$ training data-set). First, several trial points are generated along the $Y$-axis with the $X$ value as $x_{\text{query}}$. Then we classify these trial points as either in-tol or out-tol using a forward pass with both labels– 1 (in-tol) and 0 (out-tol) for all the trial points. Trial points which possess higher goodness for label 1 can be considered in-tol, and those with higher goodness for label 0 can be considered out-tol (see discussuion in 3.1.2). Next, the mean value for $y$ and the 95% confidence interval ($\pm$ twice the standard deviation) can be computed using the in-tol points. This process of function value inference is viable for functions of multiple variables as well, with $x_{\text{query}}$ being multidimensional. This process of obtaining $y_{\text{mean}}$ can be repeated over multiple $X_{\text{query}}$ points to obtain a smoother curve for the function. Algorithm 2 details function regression using a FF NN at a point $x_{\text{query}}$.

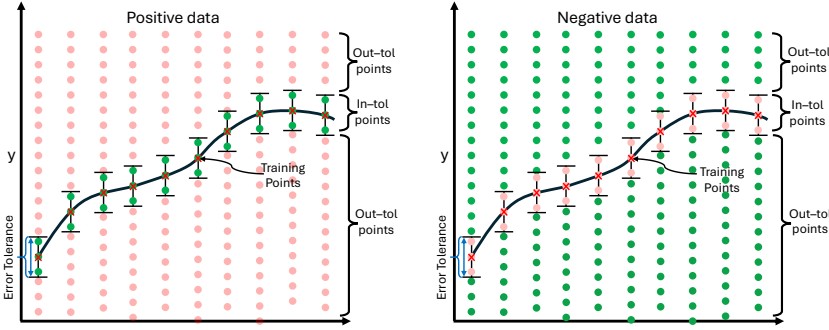

Figure 2: Schematic diagram for training of a FF NN, with the red crosses indicating the training data-points, and colored circles indicating the trial points with green corresponding to label 1.0 and red corresponding to label 0.0. On the left, the positive data has the trial points labeled correctly, while the right figure shows the negative data with incorrect labeling.

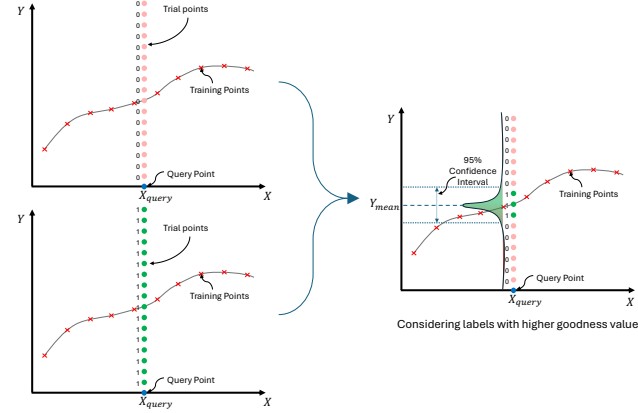

Figure 3: Schematic diagram of prediction phase while inferencing from the trained forward forward neural network. Both labels– in-tol (1.0) and out-tol (0.0) are applied to all the trial points. The label yielding the higher goodness value would ideally be chosen as the correct label for the trial point (see subsubsection 3.1.2).

---

**Algorithm 1** Forward Forward Regression Training for 1-D functions

---

**Require:**

Training dataset: $\mathcal{D}_{\text{train}} = \left\{ \left( x_{\text{actual}}^{(i)}, y_{\text{actual}}^{(i)} \right) : i \in \{1, 2, \ldots, N_{\text{actual}}\} \right\}$

Preset tolerance level: tol

1: Define a Forward Forward NN that takes 3 inputs– $x$ coordinate, $y$ coordinate and label (either 1.0 or 0.0). The NN has $N_{\text{layers}}$ number of layers, with parameters $\theta_{(i)}$ associated with the $i^{\text{th}}$ layer, and the output of the layer being $\boldsymbol{y}_{\theta_{(i)}}^{(i)}$.

2: Define arbitrary vectors $\boldsymbol{\zeta}^{(i)}$ where $i \in \{1, 2, \ldots, N_{\text{layers}}\}$.

3: Define layer-wise goodness functions $g^{(i)} \left( \boldsymbol{y}_{\theta_{(i)}}^{(i)} \right) = cos_{sim} \left( \boldsymbol{y}_{\theta_{(i)}}^{(i)}, \boldsymbol{\zeta}^{(i)} \right)$

4: Define $y_{\min}$ and $y_{\max}$ as feasible limits for the range of the function $y(x)$ in the domain of interest.

5: **for** $i = 1, \ldots, N_{\text{actual}}$ **do**

6:    $\mathcal{D}_{\text{train}} = \mathcal{D}_{\text{train}} \cup \{(x_{\text{actual}}^{(i)}, y_i^{(k)}) : k \in 1, \ldots, N_{\text{in-tol}}\}$

    the points $y_i^{(k)}$ are evenly spaced in the interval $\left[ y_{\text{actual}}^{(i)} - \text{tol}, y_{\text{actual}}^{(i)} + \text{tol} \right]$

7:    $\mathcal{D}_{\text{train}} = \mathcal{D}_{\text{train}} \cup \{(x_{\text{actual}}^{(i)}, y_i^{(k)}) : k \in 1, \ldots, N_{\text{out-tol}}\}$

    the points $y_i^{(k)}$ are evenly spaced in the interval $\left[ y_{\min}, y_{\text{actual}}^{(i)} - \text{tol} \right) \cup \left( y_{\text{actual}}^{(i)} + \text{tol}, y_{\max} \right]$

8: **end for**

9: Define correctly labelled (positive) dataset as follows:

   $\mathcal{D}_{\text{positive}} = \left\{ (x_{\text{actual}}^{(i)}, y_i, 1.0) : \left| y_{\text{actual}}^{(i)} - y_i \right| \leq \text{tol} \right\} \cup \left\{ (x_i, y_i, 0.0) : \left| y_{\text{actual}}^{(i)} - \text{tol} \right| > \text{tol} \right\}$

   Define incorrectly labelled (negative) dataset as follows:

   $\mathcal{D}_{\text{negative}} = \left\{ (x_{\text{actual}}^{(i)}, y_i, 0.0) : \left| y_{\text{actual}}^{(i)} - y_i \right| \leq \text{tol} \right\} \cup \left\{ (x_{\text{actual}}^{(i)}, y_i, 1.0) : \left| y_{\text{actual}}^{(i)} - y_i \right| > \text{tol} \right\}$

   Where $(x_{\text{actual}}^{(i)}, y_i) \in \mathcal{D}_{\text{train}} \, \forall \, i \in 1, \ldots, N_{\text{actual}}$

10: Define inputs $\boldsymbol{\xi}_{\text{pos}}^{(0)} = \mathcal{D}_{\text{positive}}$ and $\boldsymbol{\xi}_{\text{neg}}^{(0)} = \mathcal{D}_{\text{negative}}$

11: **for** i=1, \ldots, $N_{\text{layers}}$ **do**

12:    **for** epoch = $1, \ldots, N_{\text{epochs}}$ **do**

13:        Perform forward pass through layer $i$ with inputs as $\boldsymbol{\xi}_{\text{pos}}^{(i-1)}, \boldsymbol{\xi}_{\text{neg}}^{(i-1)}$ to obtain outputs $\boldsymbol{y}_{\text{pos},\theta_{(i)}}^{(i)}, \boldsymbol{y}_{\text{neg},\theta_{(i)}}^{(i)}$, respectively.

14:        Obtain $g_{\text{pos}}^{(i)}$ and $g_{\text{neg}}^{(i)}$ from $\boldsymbol{y}_{\text{pos},\theta_{(i)}}^{(i)}$ and $\boldsymbol{y}_{\text{neg},\theta_{(i)}}^{(i)}$, respectively, as explained in step 3.

15:        Compute mean $Loss^{(i)}$ across all datapoints (Refer Eq. 3)

16:        Update $\theta_{(i)}$ to minimize $Loss^{(i)}$ (Any gradient descent will do)

17:    **end for**

18:    Set $\boldsymbol{\xi}_{\text{pos}}^{(i)} = \boldsymbol{y}_{\text{pos},\theta_{(i)}}^{(i)}$ and $\boldsymbol{\xi}_{\text{neg}}^{(i)} = \boldsymbol{y}_{\text{neg},\theta_{(i)}}^{(i)}$

19: **end for**

20: **return** Final trained NN model with parameters $\theta_{(i)}$ and arbitrary vectors $\boldsymbol{\zeta}^{(i)}$ for $i = 1, \ldots, N_{\text{layers}}$

---

---

**Algorithm 2** Forward-Forward prediction of function value at $x_{\text{query}}$

---

**Require:**
   Trained FF NN model with $N_{\text{layers}}$ number of layers and similar number of arbitrary vectors $\boldsymbol{\zeta}^{(i)}$.
   The $x$-coordinate at which function ($y$) value is desired: $x_{\text{query}}$.
   An estimate of the upper ($y_{\text{max}}$) and lower ($y_{\text{min}}$) limit of the function's range Number of trial points to be generated at each query point: $N_{\text{trials}}$
1: Initialize $G_{\text{in-tol}} \leftarrow$ zeros($N_{\text{trials}}$,1)
2: Initialize $G_{\text{out-tol}} \leftarrow$ zeros($N_{\text{trials}}$,1)
3: Initialize $\xi_{\text{in-tol}}^{(0)} \leftarrow$ zeros($N_{\text{trials}}$,3)
4: Initialize $\xi_{\text{out-tol}}^{(0)} \leftarrow$ zeros($N_{\text{trials}}$,3)
5: **for** $k = 1, \ldots, N_{\text{trials}}$ **do**
6:     Define $y_{\text{trial}}^{(k)} = y_{\text{min}} + \frac{y_{\text{max}} - y_{\text{min}}}{N_{\text{trials}} - 1}(k - 1)$
7:     $\xi_{\text{in-tol}}^{(0)}[k] \leftarrow (x_{\text{query}}, y_{\text{trial}}^{(k)}, 1.0)$
8:     $\xi_{\text{out-tol}}^{(0)}[k] \leftarrow (x_{\text{query}}, y_{\text{trial}}^{(k)}, 0.0)$
9: **end for**
10: **for** $i = 1, \ldots, N_{\text{layers}}$ **do**
11:     Input $\xi_{\text{in-tol}}^{(i-1)}$ to the $i^{th}$ layer of the FF NN to obtain $y_{\theta_i, \text{in-tol}}^{(i)}$ as the output.
12:     Input $\xi_{\text{out-tol}}^{(i-1)}$ to the $i^{th}$ layer of the FF NN to obtain $y_{\theta_i, \text{out-tol}}^{(i)}$ as the output.
13:     Compute $g_{\text{in-tol}}^{(i)} \leftarrow cos_{sim}\left(y_{\theta_i, \text{in-tol}}^{(i)}, \boldsymbol{\zeta}^{(i)}\right)$
14:     Compute $g_{\text{out-tol}}^{(i)} \leftarrow cos_{sim}\left(y_{\theta_i, \text{out-tol}}^{(i)}, \boldsymbol{\zeta}^{(i)}\right)$
15:     $G_{\text{in-tol}} \leftarrow G_{\text{in-tol}} + g_{\text{in-tol}}^{(i)}$
16:     $G_{\text{out-tol}} \leftarrow G_{\text{out-tol}} + g_{\text{out-tol}}^{(i)}$
17:     $\xi_{\text{in-tol}}^{(i)} \leftarrow y_{\theta_i, \text{in-tol}}^{(i)}$
18:     $\xi_{\text{out-tol}}^{(i)} \leftarrow y_{\theta_i, \text{out-tol}}^{(i)}$
19: **end for**
20: Initialize $y \leftarrow \{\}$ (Empty set)
21: **for** $k = 1, \ldots, N_{\text{trials}}$ **do**
22:     **if** $G_{\text{out-tol}}[k] > G_{\text{in-tol}}[k]$ **then**
23:         $y \leftarrow y \cup \{y_{\text{trial}}^{(k)}\}$
24:     **end if**
25: **end for**
26: Define $y_{\text{mean}} \leftarrow mean(y)$
27: Define $y_{\text{std}} \leftarrow STD(y)$
28: **return** $y_{\text{mean}}$ and $y_{\text{std}}$ to obtain 95% confidence interval ($\pm 2y_{\text{std}}$)

---

## 3 RESULTS AND DISCUSSIONS

We validated our proposed FF regression algorithm against various benchmark 1-D, 2-D and 3-D functions. We chose functions involving combinations of the ubiquitous sinusoidal and exponential terms. Across all the regression tasks considered in the study we used a similar FF NN (dimension of input varies) with a total of 3 layers with 64, 128 and 32 neurons in each layer, respectively. We employ the GELU activation function in each layer. Further details regarding the hyperparameters employed in each benchmark is available in Table 1.

### 3.1 1-D REGRESSION

In figure 4 we provide a summary of FF-regression results for three different functions:

- $f_1(x) = sin(2\pi x) + 1$

- $f_2(x) = e^{-0.3x}cos(\frac{\pi x}{2})$

- $f_3(x) = sin(\pi x) + \frac{1}{2}cos(2\pi x)$

The plots provided show the training data points as red crosses, and the mean predicted value ($y_{\text{mean}}$) of the function as a blue curve with the shaded area denoting the 95% confidence region. Each of

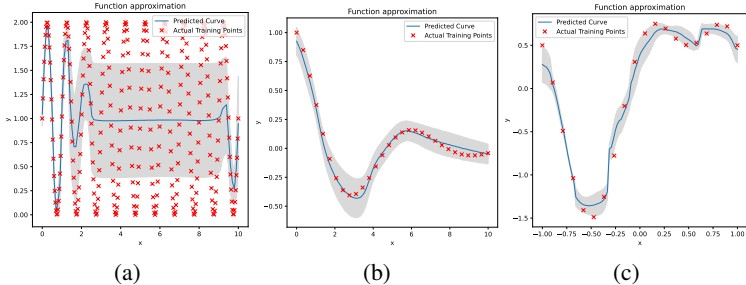

(a)                          (b)                          (c)

Figure 4: Results of FF-regrssion on 1D functions– (a) $f_1(x)$, (b) $f_2(x)$ and (c) $f_3(x)$, with the red crosses indicating the training data points, blue line indicating the mean predicted value, while the shaded area denotes the 95% confidence interval.

the plots show the predictions at their "best" with convergence evaluated after increasing the number of training data points and increasing the number of epochs of training per layer. As expected, increasing the number of training points and the number of training epochs show improvement in accuracy and reduction in uncertainty to an extent. In figure 4a we notice that FF NN is unable to approximate all cycles of the sinusoid despite a large number of training datapoints (300) and training epochs (5000), presumably for want of more complexity in the NN. Interestingly, a similar effect is observed when using Convolutional Neural Networks (CNNs), which are primarily suited to classification tasks, to perform regression tasks on periodic functions with many oscillations in the domain of interest ( Figure 11). However, the other FF NNs are able to approximate the other functions (figures 4b,4c) ,which contain 1-2 full period cycles of the function, very accurately with around 20 data points and 500 epochs of training.

### 3.1.1 COMMENT ON VARYING THE HYPERPARAMETERS

The effect of the following hyperparameters from Algorithms 1 and 2 were studied:

- tol: It was observed that decreasing the tol parameter improved the accuracy and reduced the uncertainty in the predicted results. However, too small a "tol" can result in breaks in the function prediction, wherein the entire set of trial points would be classified as out-tol. An example of this can be seen in figure 12

- $y_{\min}$ and $y_{\max}$: Figure 13 shows that as $y_{\min}$ gets too close to the least $y_{\text{actual}}$, the FF NN provides poor prediction at such points as enough number of trial points are not generated in the interval $[y_{\min}, y_{\text{actual}} - tol]$.

- $N_{\text{in-tol}}$ and $N_{\text{out-tol}}$: During training, at any given $x_{\text{actual}}$ the length of the in-tol region is clearly smaller than the length of the out-tol region. This would mean that we would need more number of out-tol points as compared to in-tol points, i.e, $N_{\text{out-tol}}$ should be significantly greater than $N_{\text{in-tol}}$. Figure 14 provides a plot of MSE vs. $N_{\text{out-tol}}$ for FF-regression of $f_3$, showing that higher $N_{\text{out-tol}}$ as compared to $N_{\text{in-tol}}$ provides more accuracy.

The hyperparameters relevant to the FF-regression for each of the functions considered in the study is summarized in Table 1. The convergence of the predicted values with the increase in the number of training epochs is demonstrated for function $f_2$ in figure 15.

### 3.1.2 COMMENT ON INVERSION OF GOODNESS

As discussed in subsection 2.5, we train the FF NN to increase its goodness value for correctly labeled data and decrease its goodness value for incorrectly labeled data. A peculiar discovery we made during inferencing from a trained FF NN is that the trial points in the vicinity of the in-tol region have a higher goodness score for the out-tol label as opposed to the in-tol label and the trial points away from the in-tol region show higher goodness scores for the in-tol label. This is in stark contrast to the training, where, as shown in figure 16, $g_{\text{pos}} > g_{\text{neg}}$. This would mean the FF NN is working in a manner exactly opposite to what was intended (which is also useful). This is the reason we use the inequality in line 22 of algorithm 2, wherein we select points with higher goodness for the out-tol label as the in-tol points.

### 3.2 2-D AND 3-D REGRESSION

In figure 5 we provide FF-regression results for the 2-D functions:

- $f_4(x_1, x_2) = x_1^2 + x_2^2$
- $f_5(x_1, x_2) = 2sin(x_1) + cos(x_2)$

We omit the uncertainty surfaces for ease of illustration, and the functions can be seen to be approximated reasonably well after 500 epochs of training per layer, with a 25×25 grid of points on the $x_1 - x_2$ plane used for training.

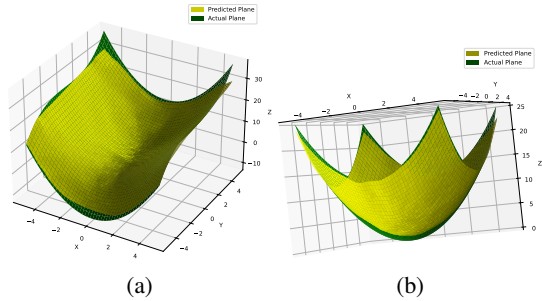

(a)    (b)

Figure 5: FF-Regression results for the 2D functions– (a) $f_4(x_1, x_2)$ and (b) $f_5(x_1, x_2)$, with the yellow surface indicating the actual function output and the green surface indicating the mean predicted function values. The training datapoints and confidence bounds are omitted for clarity.

We chose the following 3-D functions as the next benchmark for our proposed FF-regression algorithm:

- $f_6(x_1, x_2, x_3) = x_1^2 + x_2^2 + x_3^2$
- $f_7(x_1, x_2, x_3) = sin(\frac{x_1 x_2}{5}) + cos^2(\frac{x_3}{5}) + x_1 x_2 x_3$
- $f_8(x_1, x_2, x_3) = e^{\frac{x_1^2}{5}} sin(\frac{x_2 x_3}{5}) + e^{\frac{x_2^2}{5}} sin(\frac{x_1 x_3}{5}) + e^{\frac{x_3^2}{5}} sin(\frac{x_2 x_1}{5})$

We used 25×25×25 grid of training points distributed within a cubical domain with $x_1, x_2, x_3 \in [-3, 3]$. Visualizing this result would require a 4-D plot. Instead, noting that the domain of the above 3-D functions is contained within a cube, we chose to compare the true and predicted functions along certain lines in the domain. As illustrated in figures 6a, 7a and 8a, we chose the 4 body diagonals and 4 other surface diagonals on the cube to compare the true and predicted values of the functions. The FF-regression results are shown in figures 6,7 and 8, with 500 epochs of training providing satisfactory accuracy for all functions. Increasing the number of epochs further seems to marginally improve the accuracy and the uncertainty. It can be noted that the output corresponding to $f_8$ shows an unusually high uncertainty along certain lines of data. The uncertainty in $f_8$ can be expected to reduce with an increase in the number of training points.

### 3.3 BENCHMARKING AGAINST OPEN SOURCE REGRESSION DATA

We benchmark our FF regression framework on open source regression tasks (noa; Bischl et al., 2025)– i.) the Boston housing dataset (13 input features, 1 output feature), ii.) Diabetes dataset (11 input features, 1 output feature), and iii.) wine-quality dataset (11 input features and 1 output feature). We compare the Mean-Squared Error (MSE) of the FF regression framework, after 5000 epochs of training, with the MSE scores of the Random Forest regression framework that was trained and inferred for exactly the same datasets. In figures 9(a), (b) and (c), we plot the actual target value against the target values predicted from the Random Forest framework and the FF regression framework for all three benchmark datasets. It can be seen that the FF regression framework performs as well as the Random Forest regression framework in all three regression tasks, with similar MSE scores. This demonstrates the ability of the proposed FF framework to robustly handle multidimensional data.

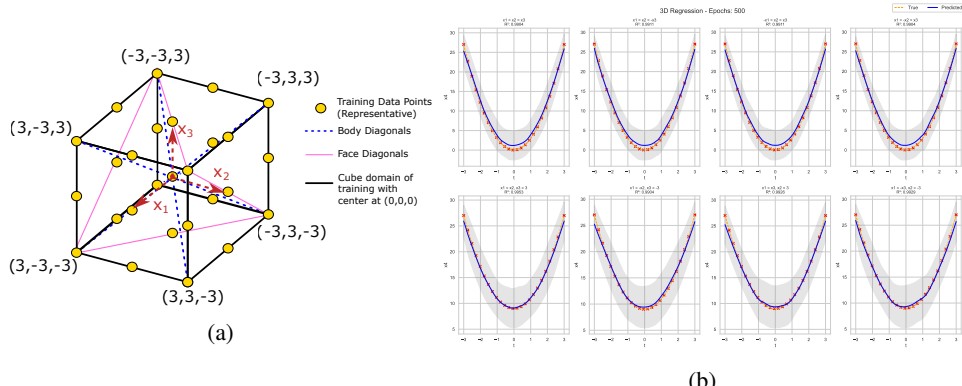

Figure 6: (a) Schematic of the domain of training of the FF NN, with the yellow spheres indicating a few of the training data points, the blue-dashed line indicating the 4 body diagonals of the cube and the pink lines indicating the 4 particular surface diagonals along which the true and predicted line plots were compared. (b) Line plots of the FF-Regression result for 3D function $f_6$, with the red crosses indicating training datapoints and the gray shading indicating the 95% confidence region.

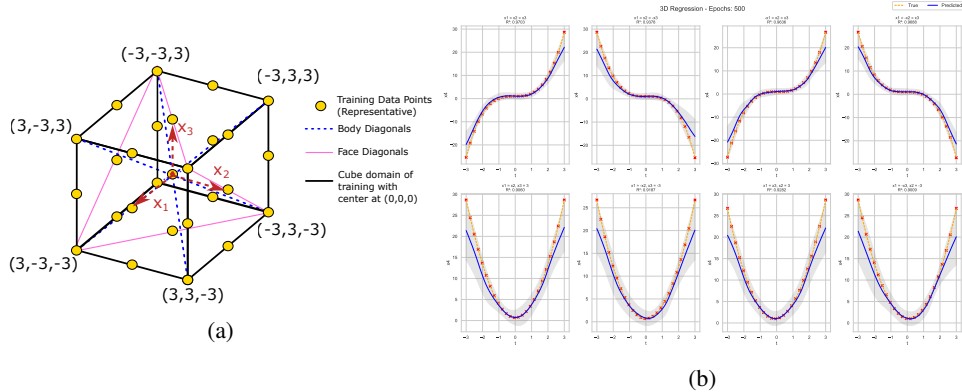

Figure 7: (a) Schematic of the domain of training of the FF NN (same as for figure 6a). (b) Line plots of the FF-Regression result for 3D function $f_7$, with the red crosses indicating training datapoints encountered along the line plot and the gray shading indicating the 95% confidence region.

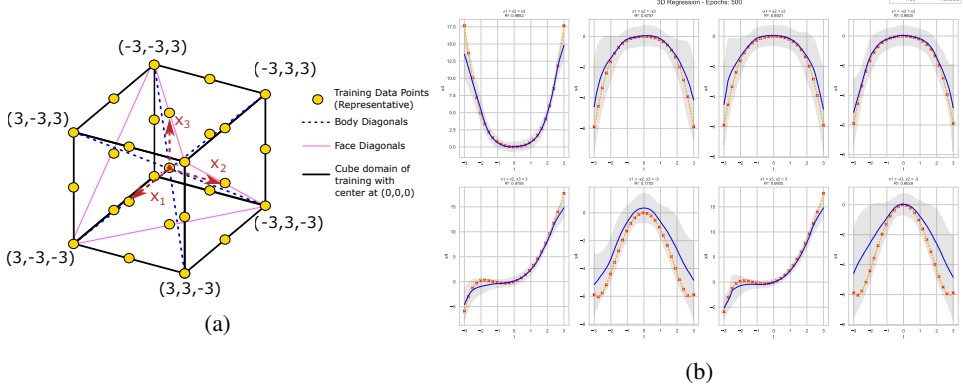

Figure 8: (a) Schematic of the domain of training of the FF NN ((same as for figure 6a)). (b) Line plots of the FF-Regression result for 3D function $f_8$, with the red crosses indicating training datapoints encountered along the line plot and the gray shading indicating the 95% confidence region.

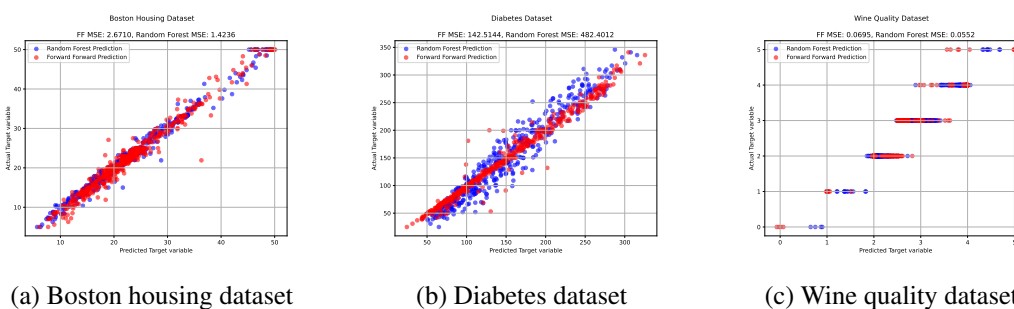

|  (a) Boston housing dataset | (b) Diabetes dataset | (c) Wine quality dataset |

Figure 9: Benchmarking FF algorithm against open source regression datasets. The actual target values are plotted against the values predicted by FF regression and Random Forest prediction. The actual target values in dataset (c) are discrete integers, whereas the predicted values are floating point numbers.

### 3.4 COMMENTS ON COMPUTATIONAL COMPLEXITY

In figure 10 we provide a plot of the total compute time for training and inferring the FF regression model for functions $f_4$ and $f_5$. It can be observed that both compute times increase almost linearly with the number of data points. This is expected, as the computational cost ($c$) associated with each data point would be the evaluation of two goodness values at exactly $N_{\text{trial}}$ number of trial points. The total computational cost $C = cN$, where $N$ is the number of data points. Thus, $C = 2N_{\text{trial}}N$, predicts a linear increase in the computational cost with the increase in the number of training and inferring data points.

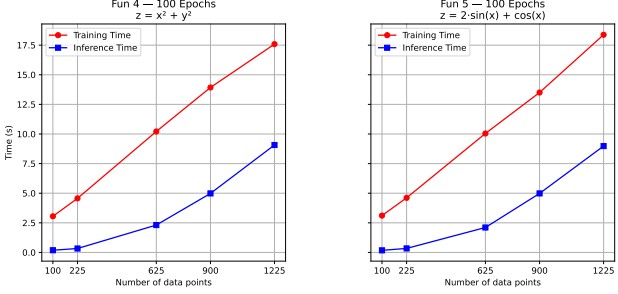

Figure 10: Variation of computation time with number of data points

### 4 CONCLUSION

In this work we proposed and validated a new algorithm for function regression using the Forward Forward method of training NNs. We successfully benchmarked the proposed algorithm against eight 1-D, 2-D and 3-D functions, and three other open source higher dimensional regression datasets. We documented the effect of various hyperparameters on the accuracy and uncertainty of the predictions. In subsubsection 3.1.2 we also noted a peculiarity wherein the trained FF NN works exactly opposite to its initial design, thereby still being able to perform function regression. We did not explore the underlying mathematical reason in this work. In Appendix B and Appendix C we provide preliminary results on extending the FF-regression algorithm to Kolmogorov Arnold Networks and Deep Physical Neural Networks, respectively. As seen in Table 2, the traditional backpropagation algorithm significantly outperforms the proposed FF-regression algorithm in terms of compute time to achieve similar accuracies. However, further studies have to be performed to ascertain if the FF-regression algorithm consumes significantly lower energy compared to BP when deployed on a fully Analog/Physical Neural Network framework.

## 5 REPRODUCIBILITY STATEMENT

We have taken all measures to ensure that all the figures and results provided in the main text and appendix are reproducible. The algorithm underlying the training and inferencing of a Forward Forward Neural Network for regression are provided in Algorithms 1 and 2 respectively. Furthermore the code to reproduce all results, alongside a README.md file with minor instructions to execute the codes is provided in the supplementary files as .zip folder. Due to the use of randomly generated vectors for evaluating cosine similarity with layer outputs, results may slightly vary from the ones presented in the main text.

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

## APPENDIX A   FIGURES AND TABLES RELATED TO THE PAPER

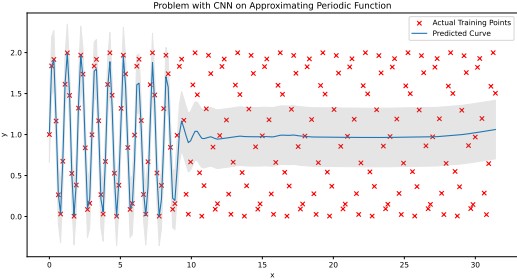

Figure 11: Performing regression using a Convolutional Neural Network for $f(x) = \sin 2\pi x + 1$. The CNN fails to predict variation in the function after a few cycles of oscillation.

| Hyperparameters | $f_1$ | $f_2$ | $f_3$ | $f_4$ | $f_5$ | $f_6$ | $f_7$ | $f_8$ |
|---|---|---|---|---|---|---|---|---|
| $N_{\text{layers}}$ | 3 | 3 | 3 | 3 | 3 | 3 | 3 | 3 |
| tol | 0.02 | 0.05 | 0.01 | 0.1 | 0.1 | 0.1 | 0.1 | 0.1 |
| $N_{\text{in-tol}}$ | 10 | 10 | 10 | 30 | 30 | 30 | 30 | 30 |
| $N_{\text{out-tol}}$ | 10 | 10 | 10 | 50 | 50 | 50 | 50 | 50 |
| $N_{\text{trials}}$ | 1000 | 1000 | 1000 | 300 | 300 | 1000 | 1000 | 1000 |
| $N_{\text{epochs}}$ | 500 | 500 | 500 | 300 | 300 | 500 | 500 | 500 |

Table 1: A list of hyperparameters used for the FF-regression of each function.

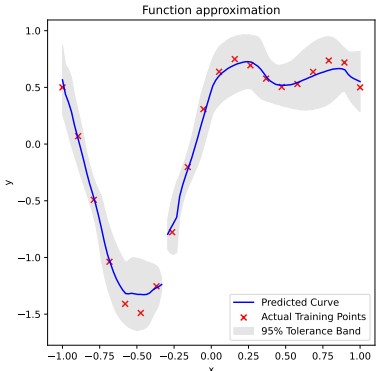

Figure 12: Prediction for function $f_3$ breaks in certain regions as we try to reduce "tol" below a certain value.

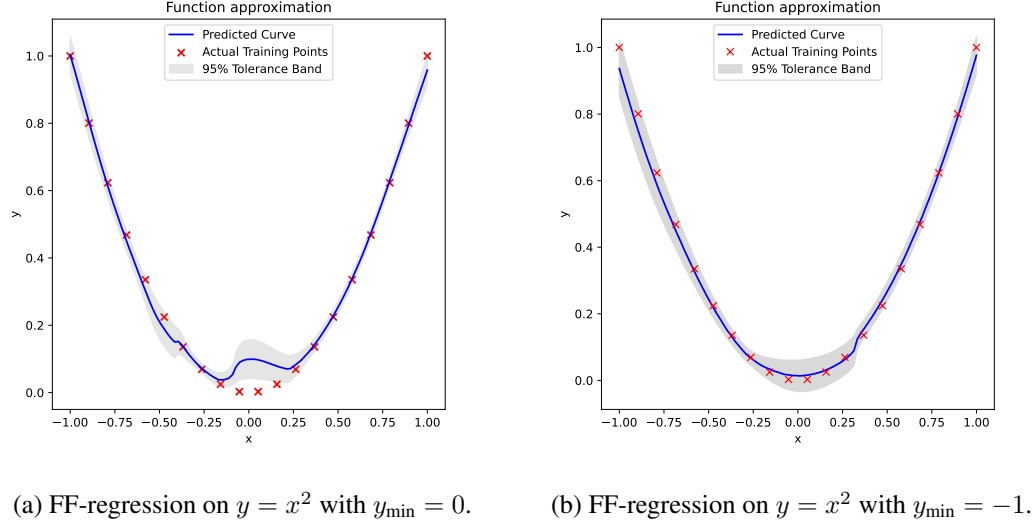

(a) FF-regression on $y = x^2$ with $y_{\min} = 0$.    (b) FF-regression on $y = x^2$ with $y_{\min} = -1$.

Figure 13: Comparative study showing that if either $y_{\min}$ or $y_{\max}$ are too close to the training data-point value ($y_{\text{actual}}$), the FF NN provides poor predictions at such points.

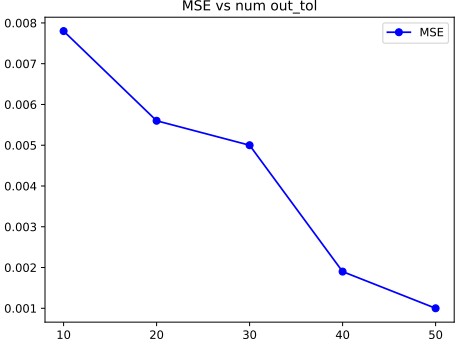

Figure 14: Plot of MSE for FF-regression of $f_3$ as function of $N_{\text{out-tol}}$ used during training.

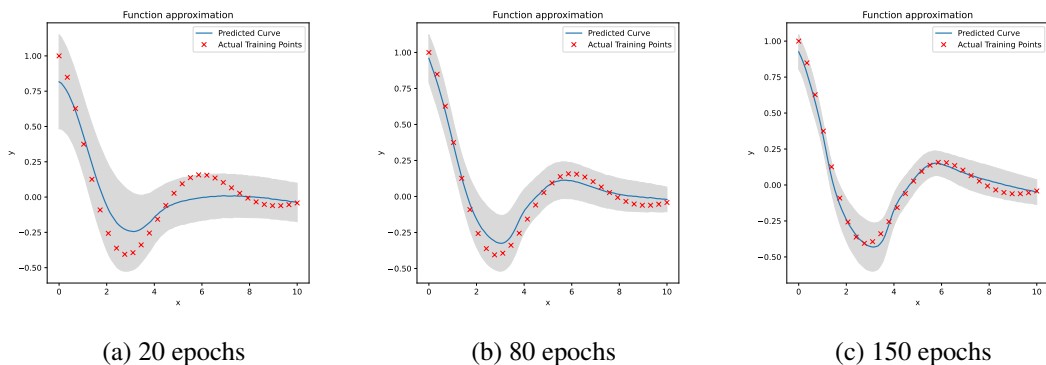

(a) 20 epochs        (b) 80 epochs        (c) 150 epochs

Figure 15: Convergence plots for function $f_2$ for $N = 30$ number of training points and increasing number of training epochs– 20, 80, and 150.

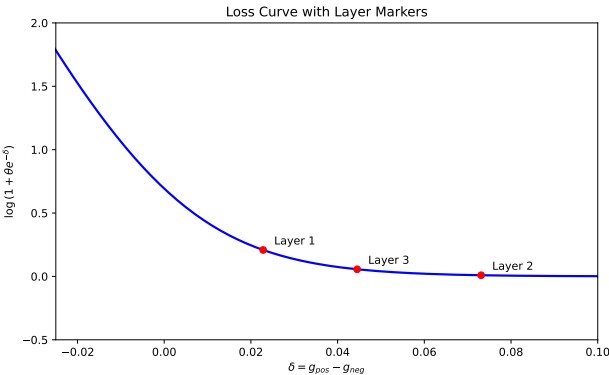

Figure 16: Plot of loss function (Eq. 3) w.r.t $(g_{pos} - g_{neg})$ for layer 1, layer 2 and layer 3 after training for $f_3$.

## APPENDIX B    ATTEMPTS WITH KOLMOGOROV ARNOLD NETWORKS (KANS)

Kolmogorov Arnold Networks are powerful in approximating complex functions with relatively fewer parameters compared to NNs. Each node of the KAN provides a spline-based approximation and layers of this network act as composite functions. We employed the proposed FF-regression algorithm to train and infer a 3-layered KAN to approximate a simple sinusoidal function $y = 2 + sin(2\pi x)$, with the output of the nodes in each layer of the KAN being used to compute the goodness of the layer. The results after 5000 epochs of training are shown in figure 17. This preliminary result seems somewhat encouraging and further studies could provide more insights on effectiveness of training and infering KANs using the FF-regression approach. (Refer to code in supplementary material for implementation details)

## APPENDIX C    RESULTS AND DISCUSSIONS FOR FUNCTION REGRESSION USING A DEEP PHYSICAL NEURAL NETWORK

We considered a 3 layered DPNN, wherein the trainable parameters are included as part of the input, and the activation function associated with each "physical layer" is $sin(x) + cos(x)$. We trained a DPNN for regression using the BP algorithm and another DPNN using the FF algorithm. A schematic of the architecture for both can be seen in figure 18.

The results for the BP and FF-based regression for the simple function $y = x^2$ can be seen in figures 19 and 20, respectively. While the BP-based regression for DPNNs provide satisfactory convergence after around 15000 epochs, the FF-based DPNN provides no semblance of convergence after 10000

| | FF Algorithm n_epochs = 500 | Backpropagation n_epochs = 500 | FF Algorithm n_epochs = 5000 | Backpropagation n_epochs = 5000 |
|---|---|---|---|---|
| $f_3$ | 6.62 s | 0.5 s | 42.34 s | 4.67 s |
| $f_6$ | 173.11 s | 0.59 s | 2519.19 s | 5.01 s |
| $f_7$ | 155.31 s | 0.49 s | 1264.66 s | 5.10 s |
| $f_8$ | 305.27 s | 0.52 s | 2534.62 s | 5.15 s |

Table 2: Comparison of compute time (training and inference) for NNs with similar number of parameters using BP and FF, for regression of various functions, using a workstation equipped with NVIDIA RTX 5000 Ada Generation, an Intel Xeon w5-2565X CPU (18 cores, 36 threads), and 128 GB of RAM.

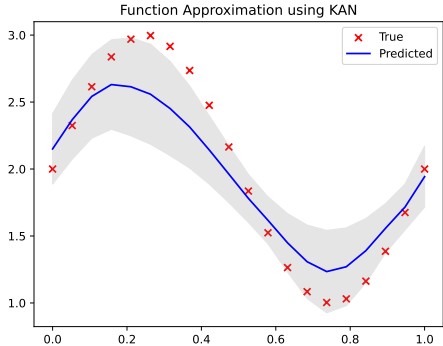

Figure 17: Forward Forward Regression implemented using Kolmogorov Arnold Networks(KANs)

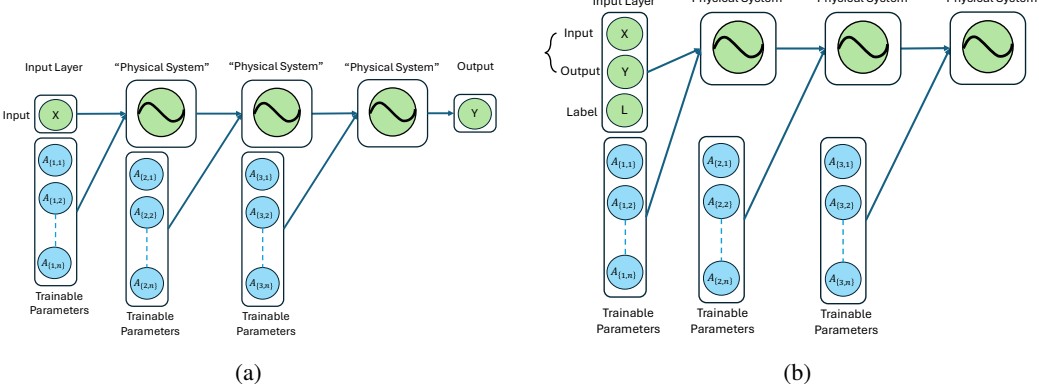

(a)                                                     (b)

Figure 18: Comparison of Deep Physical Neural Networks trained with (a) backpropagation and (b) forward-forward algorithm.

epochs of training. This indicates that further studies would be required to extend the FF-regression algorithms to DPNNs effectively.

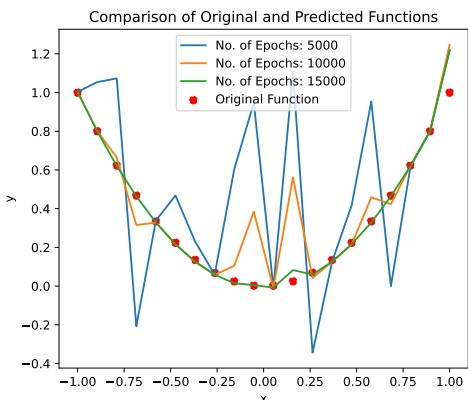

Figure 19: Result after using traditional backpropagation algorithm on physical neural networks with input layer containing trainable parameters.

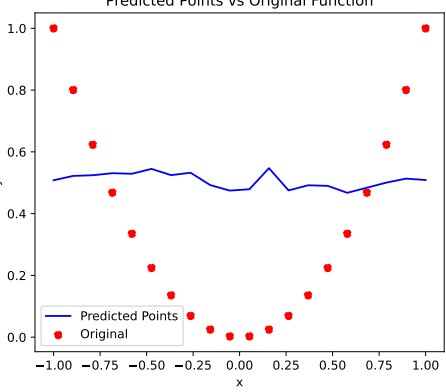

Figure 20: Result after using forward forward algorithm on physical neural networks with input layer containing trainable parameters.