# OpenReview forum: "Function regression using the forward forward training and inferring paradigm"
_ICLR.cc/2026/Conference — Submitted to ICLR 2026_

### Official Review · Reviewer_vkkW · 2025-10-31

**Soundness:** 3
**Presentation:** 3
**Contribution:** 3
**Rating:** 4
**Confidence:** 4

**Summary:**

The paper extends the concept of forward-forward networks, originally used for classification tasks, to regression tasks by carefully assigning positive and negative labels to data points. The paper tests forward-forward for regression methods on eight problems. The results look significantly good for these problems. There are no comparative evaluations with any other methods to show any significant uptake of these FF-regression for solving real-world regression problems or regression benchmarks.

**Strengths:**

The paper presents a clever trick for converting an FF classification model into an FF regression model.  It has been tested over 8 regression functions and shows the usefulness of the method.

**Weaknesses:**

The paper lacks comprehensiveness. For example, in the abstract and conclusion, it is mentioned that FF-regression is extended to Kolmogorov-Arnold Networks and Deep Physical Networks. These have not been discussed in the main paper at all. Only two Figures have been described in the Appendix without good detail.

Methods lack any comparison with other standard methods or analysis to confirm the real-world usability of the FF-regression to achieve the energy efficiency goal.

Trivial thing, but I am wondering whether the Appendix in the same PDF, which counts to 14 pages, violates the page limit.

**Questions:**

Are these functions evolved for metrics like R2
What is the performance of FF on the simple regression problem on the UCL repository? Whether authors tested this algorithm on those problems.

---

> ### Author Response · Authors · 2025-11-17
>
> Thank you for your comments. We believe they will help us improve the clarity and focus of our paper.
> Regarding weakness 1.) We agree that in the current manuscript we have omitted important details of how we extended the proposed FF regression approach to KANs. However, we do provide some explanation of using FF regression with Physical Neural Networks (PNNs) in section C of the appendix. While the intended salient contribution of our paper was the FF regression method, we included its extension to the popular KANs function regression framework to highlight our framework's relevance and applicability. We also included a preliminary study evaluating the FF regression approach's performance with Physical Neural Networks to highlight its performance shortcomings that would need to be addressed before it can be deployed to physical systems. That said, we will revise the manuscript to ensure that these "extension-studies" on KANs and PNNs are not highlighted disproportionately in comparison to the more rigorously evaluated FF regression framework. We will try to include some more relevant details on KANs and PNNs in the appendix of the manuscript. However, given the page constraints and the vast nature of each topic we will likely restrict ourselves to a few details of these "side"-studies.
> Regarding weakness 2.) We agree that we have not benchmarked our proposed FF regression approach against complicated regression tasks. We intend to do so in the revised manuscript.
> Regarding weakness 3.) ICLR guidelines allow 9 pages of main content, and stipulates no limit on the additional appendices. The revised manuscript is allowed to have 10 pages of main content and no limit on the appendices.
> Regarding question 1.) Thank you for your helpful suggestion. We will be using R^2 metric on all the function benchmarks. For a more complicated benchmark for regression tasks, we intend to use the Forest-fire area regression task provided on UC-Irvine's database (We were unable to find any by UCL). We will benchmark our proposed FF regression framework on the UC-Irvine's regression problem. We would be grateful for other suggestions regarding benchmarks to explore.

---

> ### Author Response · Authors · 2025-12-03
> **Intimation regarding changes in the manuscript**
>
> We have made the following changes in the manuscript to address the weaknesses that you highlight:
> 1.) We have removed mentions of deep physical neural networks and KANs in the abstract of the manuscript to ensure that the development and validation of the Forward Forward Regression is highlighted as the salient contribution of the paper.
> 2.)We have successfully benchmarked the FF regression approach on complex open source datasets (Boston housing dataset, Diabetes dataset and Wine-quality dataset) and compared our approach with the Random Forest approach for function regression. These results and discussions are provided in section 3.3 of the revised manuscript.
>
> We have also ensured that FF regression's performance is evaluated by the $R^2$ metric for functions f_1 to f_8 (noiseless) and the MSE metric for the open source dataset benchmarking studies (noisy data).

---

### Official Review · Reviewer_a3DU · 2025-11-01

**Soundness:** 3
**Presentation:** 3
**Contribution:** 2
**Rating:** 4
**Confidence:** 4

**Summary:**

The paper extends the Forward-Forward (FF) algorithm for classification to function regression. It treats regression as binary classification: points within a tolerance of the true value are "in-tol" (label 1), others are "out-tol" (label 0). The network uses cosine similarity as the metric to distinguish between correct and incorrect labels.

**Strengths:**

1. This is the first to apply the forward-forward (FF) learning algorithm to regression. It effectively extends a classification-only method to continuous function approximation. This fills a clear gap in the FF literature.
2. The pseudocode is clear, and the algorithms are easy to understand. Algorithms 1 and 2 provide step-by-step, reproducible training and inference procedures. The accompanying figures further enhance clarity.
3. The method successfully works on regressing low-dimensional functions. It produces reasonable approximations with meaningful uncertainty estimates on 1D, 2D, and 3D benchmarks. Results are well-visualized and supported by MSE metrics.

**Weaknesses:**

1. The method is tested only on simple, low-frequency functions, not on complex cases (high-dimensional, non-smooth, or multi-frequency). Such limitations in computational complexity are critical. Focusing on KANs may not be appropriate, because the uniform grids of KAN raise these core issues. I suggest that the author read some recent MLP variants that have solved these problems using the multi-scale mesh (a standard data structure in the finite element method).
2. The approach is low-efficient, with training and inference times orders of magnitude slower than backpropagation (Table 2). Despite avoiding backpropagation, the method still relies on gradient descent with no acceleration in convergence (Algorithm 1). Scaling-law analysis would likely reveal poor sample and computation efficiency.
3. The evaluation metrics are inconsistent: $R^2$ score should be used for noise-free regression, and MSE for noisy data. No comparison is made with standard mathematical or ML regressors (e.g., splines, tree-based models). The paper compares only with physical Neural Networks, which is misleading. Neural networks primarily focus on classification tasks (e.g., image generation, language modeling) and are poor at highly accurate regression tasks.

**Questions:**

Suggestions for Improvement

1. Broader Experiments: Test on more datasets or higher-dimensional/multi-frequency functions. Compare with traditional mathematical/ML methods.
2. Enhancements to Method: The author can explore using the multi-scale mesh instead of the uniform grid on neural networks, which is a solution to extend the application scope to higher-dimensional/multi-frequency functions and even challenge recognition tasks (such as image classification on ImageNet )

---

> ### Author Response · Authors · 2025-11-17
>
> We are grateful for your helpful comments. We believe they will help us to improve the quality of our manuscript.
> Regarding weakness 1.) We agree that we benchmark our proposed FF regression framework on simple functions. In the revised manuscript, we intend to evaluate the performance of the proposed approach to multidimensional regression benchmarks, such as the forest-fire area problem on the UC-Irvine repository. That said, in the current manuscript, we demonstrate that the proposed FF-regression approach fails to perform well for f1(x), which involves many cycles of oscillations of the sin(x) function. While our framework performs well to approximate 1 or 2 cycles of oscillations, it fails to perform well for a large number of cycles of oscillations. We believe this might be due to an inherent drawback in extending simple classification frameworks to regression problems, as even CNNs used for regression tasks face the same issue of not being able to approximate functions containing a significant number of oscillation cycles. However, in the case of CNNs, the regression capabilities can be improved by performing convolutions in the Fourier domain (as was done in Fourier Neural Operators). A similar extension might be needed to improve FF-regression's capabilities, which would perhaps be explored in a future work.
> Regarding weakness 2.) We agree that the proposed approach is indeed computationally more expensive compared to the traditional backpropagation approach as seen Table 2. In the revised manuscript, we intend to provide a brief discussion on the computational complexity of the proposed approach. However, a core point of the proposed method is its ability to be deployed on analog/physical systems, where the energy efficiency gains might outweigh the increased computational complexity. Of course, only a physical implementation of the proposed approach can provide a definitive verdict regarding its utility and superiority/inferiority compared to the backpropagation-based approach. That said, the proposed approach at least provides a fully-analog framework that can be evaluated in the future. To the best of our knowledge, no other similar analog framework for regression exists.
> Additionally, you raise an excellent point regarding the need of gradient-based optimization for each layer. Theoretically, it would still be possible to use gradient-free optimization techniques, which would of course have slower convergence, but would be more feasible for deployment on analog systems. Again, the energy efficiency gains by the analog framework might outweigh the increased computational complexity of the proposed approach compared to backpropagation.
> Regarding weakness 3.) Thank you for excellent point regarding the inappropriate use of the MSE metric. We will make use of the R^2 metric for all functions in the revised version of the manuscript. The later part of your comment in this regard was a bit unclear. We do not compare the proposed approach with physical neural networks anywhere. We only implement it in physical neural networks in the appendix in figure 16. However, we will compare the MSE metric of our proposed approach against that of other regressors - like GPR, on benchmark regression tasks - such as the Forest-fire area regression task on the UC-Irvine data repository. We would also be grateful for any other suggestions regarding regression benchmarks to test our framework.
>
> Regarding question 1.) As mentioned above, we will try to benchmark our approach on open-source regression tasks.
> Regarding question 2.) We would like to refrain from exploring multi-scale meshed- methods in the current work, as we don't see how such an approach could retain the potential of being deployed on physical neural networks. Additionally, the uniform grid enables a parallelized and vectorized evaluation of the goodnesses at each trial point. In contrast, having adaptive sizing of the grid at each query point would likely serialize the evaluation leading to much higher compute times.

---

> ### Author Response · Authors · 2025-12-03
> **Intimation regarding changes in manuscript**
>
> We have made the following changes to the manuscript to address the weaknesses that you highlight:
> 1.) We have successfully benchmarked our FF regression approach to complicated higher dimensional regression tasks on open source datasets. These results and discussion are provided in section 3.3. Further, in lines 343 to 345 we also provide a discussion on  a similar inability of CNNs to accurately perform regression tasks on datasets that contain many cycles of oscillations, with a supporting example in figure 11.
> 2.) We have commented on the computational complexity of the proposed algorithm in section 3.4 of the revised manuscript.
> 3.) We have now changed all accuracy estimation metrics for functions f_1 to f_8 to R^2 instead of the MSE metric used earlier. However for the higher dimensional open source data regression tasks, we still use the MSE metric to account for noise in the data. We thank you for this input.
>
> We would also like to reiterate that our framework (to the best of our knowledge) is the first to use the Forward Forward method to perform regression tasks and successfully benchmark it using various studies. We believe that this forms an important foundation to assess means to implement physical/analog neural networks in the future.

---

### Official Review · Reviewer_ZMCD · 2025-11-03

**Soundness:** 2
**Presentation:** 3
**Contribution:** 2
**Rating:** 4
**Confidence:** 4

**Summary:**

This paper introduces the forward-forward (FF) framework for training and performing inference with regression neural networks. The proposed FF algorithm adapts the FF approach originally designed for neural classifiers to regression tasks.

Similar to its use in classification, the FF framework for regression relies on positive data, negative data, and a goodness function. The algorithm reframes regression as a classification problem by creating bins of target values based on training data and a user-defined tolerance level. If a bin contains the true value y for a given input  x, the algorithm assigns a label of 1 (positive data); otherwise, it assigns a label of 0 (negative data). This transformation enables binary classification between positive and negative data.

During inference, the FF framework maps a query input x to its predicted value. It generates trial points to define candidate bins, identifies the bins where x qualifies as positive data, and collects the corresponding trial points labeled as 1. The algorithm then computes the mean of these trial points to produce the final prediction.

**Strengths:**

The paper presents its contribution clearly and organizes its content effectively. The introduction and theoretical background offer relevant context  that positions the work and highlights its contribution. The presentation of the forward-forward approach for neural classifiers helps with understanding the new regression framework as a reader. The suitability of the forward-forward algorithm for the implementation of neuromorphic computing and physical analogs for neural networks underscores the significance of this work.

**Weaknesses:**

The quality of the experimental result in the paper is low and this undermines the soundness and utility of this paper.

1)  Experimental results for the hyperparameters: This paper does not present experimental results that illustrates the effect of the hyperparameters (tol, $y_{min}$, $y_{max}$)) on the performance of this method. Subsection 3.1.1 summarizes the effects but fails to provide quantitative evidence to support this.

2). Limited simulations: The simulations in this paper is restricted to datapoints from a closes form expressions. It is unclear if the performance on closed form expressions with translate to typical ML-based regression problems, where you only have access to input-output pairs and not a closed-form expression.

3). No information about the computational cost: For the inference, the new methods uses trial points to define candidate bins. The inference maps a query input x to the bins that result from these trial points. It is important to understand the computational cost of this approach. In the classification case, the number of runs per inference is K where K is the number of classes or categories. For regression, it seems that the number of runs depends on the number of trial point. But there is no information about the impact on the computational cost.

**Questions:**

Please refer to the points under weaknesses.

I also noticed a typographical error on line 183: " ... to obtain a the mean and the standard deviation..."

---

> ### Author Response · Authors · 2025-11-17
>
> Thank you for your helpful comments. We hope to improve the quality of our manuscript in line with these comments.
> Regarding weakness 1.) We had attempted to provide a qualitative demonstration of the importance of choosing the right tol, y_min and y_max in Figures 9 and 10 in the appendix. In the revised manuscript, however, we will try to provide a more clear quantitative estimate of the choice of right hyperparameters for FF regression.
> Regarding weakness 2.) In the revised manuscript we intend to benchmark our algorithm against a few open source regression datasets (like the one on forest fires area by UC-Irvine). We believe this will adequately address this concern of evaluating the proposed FF regression on non-closed form datasets.
> Regarding weakness 3.) In the revised manuscript we intend to provide a section describing the computational complexity of the proposed approach.
> Thank you for pointing out the typo in line 183. We will rectify it in the revised manuscript.

---

> ### Author Response · Authors · 2025-12-03
> **Intimation regarding changes in the manuscript**
>
> We have made the following changes to the manuscript to address the weaknesses that you pointed out:
> Weakness 2.) In section 3.3 of the revised manuscript we demonstrate that the proposed FF regression framework can be easily applied to regression tasks on high dimensional datasets with only input-output pairs provided. We benchmarked our framework on i.) Boston housing dataset, ii.) Diabetes dataset, iii.) Wine-quality prediction dataset, and compared our frameworks performance with the Random Forest regression approach. It turns out the FF regression achieves similar accuracy to the Random Forest approach, with good MSE scores.
>
> Weakness 3.) In section 3.4 of the revised manuscript we comment on the computational complexity of the proposed approach. We observe that the computational complexity of the FF regression framework increases linearly with an increase in the number of datapoints for a fixed number of trial points at each data point. Thank you for your inputs in this regard.
>
> We have also corrected the typo you mentioned.

---

### Official Review · Reviewer_43M2 · 2025-11-04

**Soundness:** 4
**Presentation:** 2
**Contribution:** 3
**Rating:** 4
**Confidence:** 4

**Summary:**

This paper introduces a new methodology for approximating functions, that is, function regression, using the Forward-Forward algorithm.

**Strengths:**

1. The studied topic focusing on function regression with FFA is interesting.

2. The algorithms are concrete, enabling the reproducibility and contributing to understanding of the proposed method.

**Weaknesses:**

1. The experiments are almost conducted in a toy environment. For example, the target functions approximated here are of few structures, including only elementary functions and their linear combinations. Such regression tasks are extremely simple for neural networks.

2. This paper does not discuss the computational complexity of the method, let alone how to implement it on dedicated hardware. In the main text and experiments, the authors only demonstrate the regression results. The data examples provided offer very limited support for the overall rationale of the method. The authors do not report the runtime or convergence.

**Questions:**

1. The function regression in Figure 4(a) does not perform well. I am concerned about the relationship between the training data and the objective function here.

2. What are the challenges in extending this method to multiple dimensions? Is it necessary to design a goodness function or positive-negative data on a case-by-case basis?

---

> ### Author Response · Authors · 2025-11-17
>
> Thank you for the helpful comments.
> Regarding weakness 1.) We will be performing a regression benchmarking against more complicated datasets- such as the UC- Irvine's Forest Fire area dataset. We would be grateful if you would have other suggestions for standard open source dataset benchmarks for function regression. We believe that this will demonstrate our framework's ability to perform more complicated regression tasks.
> Regarding weakness 2.) We will add a comment regarding the complexity of our algorithm in a revised edition of the manuscript. We will also be providing convergence studies in the appendix. We have already provided a summary of the runtimes in Table 2 in the Appendix due to space constraints.
> Regarding Question 1.) We benchmark our algorithm against function 8 specifically to demonstrate the limits of using the FF-regression approach. The FF learning is specifically designed for classification problems and cannot be expected to perform very well for function regression tasks in its current state. Simple Convolution Neural Networks face the same limitations, i.e, they do not perform regression tasks well for periodic functions containing significant number (~6-7) of full cycle oscillations. However, Fourier Neural Operators, which perform Convolutions in the Fourier Domain achieve better approximation accuracies. Similar extensions might be necessary to make our proposed FF regression effective for periodic functions. We will ensure that we provide discussions about this in the manuscript.
> Regarding Question 2.) As discussed in the results of our paper, we have already extended our approach to multiple dimensions- 2 and 3. It will involve computing the goodness function (same definition as in 1D), across significantly larger number of trial points as compared to 1D. We hope to provide more clarity about this in our discussion about computational complexity in the revised manuscript.

---

> ### Author Response · Authors · 2025-12-03
> **Intimation regarding changes in the manuscript**
>
> We have now incorporated changes in the manuscript to address the weaknesses that you highlighted.
> 1.)In section 3.3 of the revised manuscript we successfully benchmark our proposed approach for complicated higher dimensional regression tasks available from open source datasets.
> 2.)In section 3.4 of the revised manuscript, we provide a brief discussion over the computational complexity and provide training and inference time vs number of datapoints plots to corroborate our discussion. We also provide an illustration of convergence of the predicted value with increasing number of epochs in figure 15.
> Regarding Question 1.) In lines 343 to 345 of the revised manuscript we add another comment, corroborated by Figure 11 of the revised manuscript, to show that even CNNs used for regression tasks fail to capture large number of oscillations in the data, similar to the way our FF regression algorithm fails in Fig. 4(a). This could perhaps be an inherent shortcoming in extending frameworks that are designed primarily for classification to regression tasks.
> Regarding question 2.) We have successfully tested the FF regression framework on datasets containing 13 dimensional features, with the same goodness function and algorithm as was used for functions f_1 to f_8.

---

### Meta-Review · Area_Chair_nHMY · 2026-01-05

**Summary:**

All reviewers found the direction potentially interesting but collectively viewed the current evidence as insufficient for acceptance. The dominant concerns were (i) evaluation being essentially “toy” and lacking standard real-world regression benchmarks, (ii) missing or underdeveloped analysis of computational complexity/runtime/convergence (especially given the paper’s hardware/efficiency motivation), and (iii) lack of convincing comparisons to standard regression baselines.
The rebuttal/updates indicate meaningful revisions (benchmarking on open-source datasets, complexity discussion, and metric clean-up). However, the overall record still reads as “promising but not yet demonstrated” for practical utility.

**Reviewer Concerns:**

Addressed:

• Need for harder, real datasets: authors committed to (and later report) benchmarking on open-source, higher-dimensional regression datasets.

•	Complexity/runtime/convergence: authors state they will add complexity commentary and convergence studies, and later report adding a complexity discussion.

•	Evaluation metric confusion: authors report switching the synthetic-function metric to $\mathbb{R}^2$ and using MSE only for noisy real data.

•	Over-emphasis on KAN/PNN “extensions”: authors acknowledge disproportionate emphasis and plan to de-highlight / reframe those side studies.

Still outstanding:

•	Baseline comparisons remain thin relative to claims: reviewers explicitly asked for comparisons to standard methods and for evidence that the approach is useful beyond curated functions.

•	Efficiency/energy narrative is still speculative: reviewers flagged the lack of analysis confirming real-world usability toward energy-efficiency goals, and rebuttal does not yet establish that case decisively.

•	Experimental rigor gaps: e.g., limited hyperparameter/ablation-type evidence was raised and is not clearly resolved in the discussion record.

**Reviewer Scores:**

43M2: likely 4 (no change) unless the new benchmarks convincingly demonstrate non-toy relevance and include strong baselines.

vkkW: could move from 4 to 6 if credible benchmark comparisons are provided and the KAN/PNN framing is clarified; otherwise, stays at 4.

ZMCD: likely 4 (no change), as concerns emphasize experimental quality/rigor beyond what is clearly closed in the thread.

a3DU: likely 4 (no change), with core uncertainty about practical advantage/efficiency and evaluation breadth persisting.

---

### Decision · Program_Chairs · 2026-01-26

Reject